# Possibilities of Using Geopolymers in Welding Processes and Protection against High Temperatures

**DOI:** 10.3390/ma16217035

**Published:** 2023-11-03

**Authors:** Sławomir Parzych, Maja Paszkowska, Dawid Stanisz, Agnieszka Bąk, Michał Łach

**Affiliations:** 1Chair of Material Engineering and Physics, Cracow University of Technology, Jana Pawła II 37, 31-864 Cracow, Poland; slawomir.parzych@pk.edu.pl (S.P.); paszkowska.maj@gmail.com (M.P.); 2Wiśniowski Sp. z o.o. S.K.A., Wielogłowy 153, 33-311 Wielogłowy, Poland; d.stanisz@wisniowski.pl

**Keywords:** geopolymers, welding process, heat and fire resistance, high temperatures

## Abstract

Geopolymer materials have long been known for their competitive properties against traditional construction materials. Their special features include high resistance to elevated temperatures and good fire resistance. They are typically used as insulating materials at temperatures not exceeding 100 °C (because they can achieve a thermal conductivity coefficient of 0.060 W/m × K or less under these conditions). Still, they can also be used as thermal insulation at temperatures exceeding 1000 °C. One technology that uses very high temperatures is metal welding technology, where temperatures often exceed as many as 3000 °C. Geopolymers, due to their properties, can also be an interesting new alternative in various welding applications. This paper presents the preliminary results of pot-proofing the resistance of geopolymers to temperatures exceeding 3000 °C. Test results of a foamed geopolymer insulating a steel substrate are presented, and a geopolymer mold for thermite rail welding was made and realistically tested. The results confirmed the feasibility of using cast geopolymer molds for thermite welding of railroad rails. The geopolymer material performed well during the test and no cracks or other damage occurred. The following article presents the potential of using geopolymer materials for welding applications.

## 1. Introduction

Geopolymers are a relatively new group of aluminosilicate materials whose synthesis is based on alkaline activation, usually in NaOH solution. These materials have interesting properties such as high temperature resistance, low shrinkage, high mechanical strength, etc. [1,2,3]. So far, the greatest interest has been aroused by these materials in terms of their use in the construction industry (for the production of various types of building materials). Every year, thousands of different scientific articles and studies on geopolymer materials and composites are written around the world. Attempts are being made to use them in more and more new applications. Due to some advantages over other materials, it seems that the possibilities for their application are virtually unlimited. The ability to synthesize these materials and obtain different properties by controlling the type of precursors for geopolymerization as well as the type of alkali activators and molar ratios makes it possible to create a geopolymer dedicated to specific applications. Due to their resistance to high temperatures [4,5,6], geopolymers can prove to be a very attractive material for welding applications.

The scientific community’s interest in geopolymers and their insulating properties and fire resistance has been particularly evident in recent years. There has been a noticeable increase in the number of publications on this topic, especially in the last 10 years. Figure 1 presents the number of scientific articles registered in the SCOPUS database according to the year of publication. Undoubtedly, geopolymer materials are an attractive material for fire protection applications and their use for this purpose will be increasingly researched and developed. It can be assumed that these materials will also be increasingly used in welding processes. It is also noticeable that the greatest interest in geopolymers as insulating materials [7,8,9] is observed in countries where high average annual temperatures are recorded. A summary of the countries in which scientists most often undertake research on geopolymers for thermal and fire insulation is shown in Figure 2.

Fire protection can be divided into passive protection and active protection [10]. Active protection includes, for example, sprinklers that are activated by a specific stimulus, most often temperature. Passive protection, on the other hand, is material solutions in the form of foams or coatings, the main purpose of which is to slow the spread of fire and temperature and minimize the effect of temperature on objects that should be protected. The solutions currently in use meet the tasks set for them to varying degrees, which is why this field is constantly being developed and newer improvements are being made to existing methods. Brand-new innovative materials and technologies that could provide a much higher level of fire safety are also being researched. Geopolymer foams are precisely one such alternative to the materials currently in use [11,12,13].

Since geopolymers are characterized by very high fire resistance, they can also be used to protect surfaces and steel structures. However, it is important to keep in mind that they can play a much more important role in the construction industry as fireproofing and structural protection materials. Geopolymers as alternative binder systems are gaining more and more interest and can exhibit unique technical properties such as high strength, high acid resistance, or high temperature resistance [14,15,16]. Compared to Portland cement-based concretes, geopolymers maintain their strength properties under temperature. After exceeding 450 °C, conventional concretes have a very large decrease in strength properties due to crushing, which in turn is the cause of their inability to evaporate water fast enough, and the resulting pressure damages their structure. Geopolymers have far more internal interconnected micropores that facilitate the passage of water vapor. When heated to 800 °C, the volume of the pores increases by 26%, and when they reach 1000 °C, they increase by a further 3%. Thanks to such properties, geopolymers can, for example, find applications in the construction industry at junctions with windows and doors and as components of fireproof elevator doors and many others [11,17].

Thanks to the high fire resistance of geopolymers, it is also possible to use them to protect steel surfaces. Geopolymers have good adhesion to steel of various types and can successfully be used for protective coatings [18,19,20]. One of the methods of producing protective coatings on steel, especially ceramic ones, is the plasma spraying method. Plasma-sprayed coatings provide metal or non-metal surfaces with increased corrosion and abrasion resistance, greater hardness, protection against the erosive action of liquid metals, low thermal conductivity to protect against thermal shock, etc. Plasma spraying technology is used to produce objects of any shape, difficult to manufacture by other technologies. Rocket nozzles, missiles and missile protection cones, crucibles, dies, tools, and Pelton scoops of ceramic and metal materials are produced in this way. According to the literature [21,22], plasma spraying technology involves melting a material in powder form with the heat of a plasma arc and throwing a stream of plasma gas at the particles melted in it onto the sprayed surface. Powder particles accelerated by the plasma jet strike the rough, cleaned surface, firmly adhering and jamming into irregularities. A mechanical, adhesive, or metallic diffusion bond can occur between the coating and the substrate material. The type of this bond depends on the presubstrate material, technological conditions, and the type of coating material. All metallic and cermet materials and most ceramic materials such as oxides, carbides, and silicides are listed as materials used for plasma spraying. Coatings made of borides and nitrides as well as thermoplastics are also sprayed. The most common are coatings made of Al_2_O_3_, Al_2_O_3_ + TiO_2_, ZrO_2_, WC, Cr_3_C_2_, and Cr_2_O_3_. Geopolymers can also be a potential material for plasma-sprayed protective coatings. Due to their chemical composition and the form in which they occur, they can form a durable ceramic coating. Their melting temperature is also suitable for use in plasma applications.

The literature also reports the possibility of using geopolymers as ceramic primers for welding stainless steels. Paper [23] presents the thermal resistance of a ceramic primer made of a metakaolin-based geopolymer for welding stainless steel. Geopolymer ceramic primers were designed in three types. The results showed that geopolymer ceramic primers can be used in welding stainless steels.

Since geopolymers are similar to ceramic materials and can carry certain loads even at very high temperatures, there is reason to believe that they can also be useful for making molds for thermite welding of railroad rails. The operation, maintenance, and construction of contactless tracks requires the use of professional technologies and materials, including processes that enable the permanent welding of rails. One method commonly used is thermite welding of rails [24]. Thermite is a mixture of iron oxide and powdered aluminum. When thermite is heated to 1000 °C, the iron oxide reduction reaction begins. Thermite burns with high heat emission and contributes to the appearance of liquid iron, which fills the molds to form a permanent weld [25]. In thermite welding, molds and the material from which they are made are extremely important, since the molds are responsible for the quality of the weld made. Welding molds must be stored in the same way as termite portions: in dry rooms, protected from moisture and damage, and in winter conditions—in a heated place at a temperature of at least +15 °C. Welding molds that are soggy must not be used even if they have been dried. It is forbidden to use damaged welding molds. If damaged molds are used, the weld will come out non-conforming and will need further repair [26].

The results presented in this article represent an innovative approach to the topic of using geopolymer materials for welding applications. So far, no similar idea of replacing ceramic molds with geopolymer molds has been proposed. The presented results of preliminary tests can contribute to the development of this issue and refine the technology of using geopolymer materials in welding. This article presents only selected results of preliminary tests confirming the very high thermal resistance of geopolymers, ease of molding (casting), and suitability of such materials as an innovative alternative to commonly used ceramic molds.

This work presents only preliminary results of using geopolymers as cladding in the process of thermite welding. This work did not focus on testing the quality of the welds obtained; instead, the purpose of this work was to confirm the suitability of geopolymers for a specific welding method, such as thermite welding. The test results confirmed that geopolymers can successfully replace the currently used ceramic molds/claddings. So far, geopolymers have not been used for this type of components and the application of geopolymers in welding processes is negligible. The presented solution is an innovation on a global scale and can contribute to the development of the use of geopolymers in high-temperature industrial processes, including, in particular, metal joining processes. This article also presents the results of studies on the thermal resistance of geopolymers, as well as the insulating properties of foamed geopolymers.

## 2. Materials and Methods

Fly ash from the Skawina Combined Heat and Power Plant (Skawina, Poland) was used to produce geopolymer foams. XRF oxide analysis of the fly ash is shown in Table 1.

Foamed geopolymers were made by mixing fly ash and sand in a ratio of 1:1 and adding an alkaline solution (14 M NaOH + aqueous solution of sodium silicate) in the amount of 400 mL per 1 kg of bulk ingredients. This mix was stirred uniformly for 10 min and then 3% hydrogen peroxide H_2_O_2_ was added, after which it was immediately transferred to molds and, without using compaction on a vibrating table, moved to cure in a laboratory dryer. Hydrogen peroxide was added as a blowing agent.

Geopolymers for rail thermite welding tests were produced using a 10 M aqueous solution of NaOH (99% purity) along with an aqueous solution of sodium silicate (1:2 ratio). The geopolymer composition also contained construction sand in a 1:1 ratio (sand/fly ash). Technical sodium hydroxide flakes and an aqueous solution of sodium silicate R-145 with a molar modulus of 2.5 and a density of about 1.45 g/cm^3^ were used to make geopolymer masses. The added batch of water was “mains” water; distilled water was not used. The alkaline solution was prepared as follows: solid sodium hydroxide was poured over an aqueous solution of sodium silicate and water. The solution was stirred thoroughly and left until the concentration equilibrated and reached a constant temperature. The solid ingredients, i.e., fly ash and sand, were mixed dry until a homogeneous mixture was obtained; then, the alkaline solution was added and mixed thoroughly. Mixing was carried out in a laboratory mixer (LMB-s; standard mixer according to PN-EN 196-1:2016-07 [27]) for about 15 min. After obtaining a homogeneous mass of dense plastic consistency, the mixtures were transferred to previously prepared silicone molds, which were then vibrated on a vibrating table.

Foamed geopolymers were produced according to the instructions described above (as for solid materials), with the exception that 15% ash microspheres were introduced into the mixture, and hydrogen peroxide H_2_O_2_ 36%, in the amount of 3%, was used after obtaining a dense plastic consistency.

Thermite mold casting was carried out using Silicone 128 PU/2, a silicone for ceramic casting. The silicone was weighed and cast in 1 kg increments to gradually flood the mold. For each kilogram of silicone, 5% of hardener was added to each pouring step, according to the manufacturer’s instructions and recommendations. The molded geopolymer concretes were annealed in a laboratory dryer for 24 h at 75 °C at atmospheric pressure. After 24 h, the specimens were pulled out and unmolded. The unmolded geopolymer shapes were then glued together with a special thermal adhesive which is resistant to high temperatures up to 1200 °C. The operation was repeated four times, making four geopolymer molds for thermite welding. The samples were made by casting the geopolymers into a self-made silicone mold. This mold was created by replicating the thermite welding mold currently used by PKP S.A.

Figure 3, Figure 4 and Figure 5 below present the stages of mold making: prepared models to be flooded in silicone for mapping—Figure 3; models flooded with silicone and molded—Figure 4; finished geopolymer molds after gluing into the target shape—Figure 5.

At the stage of pouring the geopolymer mass, the geopolymer molds were longitudinally reinforced with a 6 mm diameter steel wire, using two reinforcing wires for each half of the mold. Since it was not possible to create a one-piece casting mold for geopolymers, it was decided to cut the model to be reproduced and make a two-piece mold. After the castings were made, it was necessary to glue both molds together.

This article introduces the SoWoS welding method for thermite welding of rails. It is a method of welding with top preheating, without linting the ends of the rails. Thermite welding is based on the permanent joining of encased rails with a refractory mold, using a liquid filler material obtained during the thermionic reaction that occurs in the crucible. The rails are heated with a torch and the welding binder is poured into the welding mold. After it solidifies, the lint is cut off. The joining area is then ground [25].

Thermal conductivity, λ, was tested on an HFM 446 plate apparatus. The apparatus was based on ASTM C518 [28], ASTM C1784 [29], PN-ISO 8301 [30], JIS A1412 [31], PN-EN 12667 [32], and PN-EN 12664 [33]. It had a conductivity range of 0.007 to 2.0 W/m × K, an accuracy of ±1–2%, a repeatability of ±0.25%, and a reproducibility of ±0.5%. Temperature control and regulation were verified using a Peltier system. Thermal parameters were determined using the above device with the hot and cold plate method.

Strength tests were conducted on a MATEST 3000 testing machine and a measuring range of up to 300 kN. The test was used to conduct compressive strength tests, which confirmed the mechanical strength of the specimens. The speed of the test was 10 mm/min. In the building and construction sector, the document that regulates the method for determining the compressive strength of concrete specimens is PN-EN 12390-3:2019-07 (Testing of concrete—Part 3: Compressive strength of test specimens) [34]. The shape and dimensions of the elements to be tested are regulated by PN-EN 12390-7:2013-03 [35]. The standard shape for compression test specimens is a cube or cylinder. Compression testing involves loading the specimens until a critical value that will cause the material to fail is reached. Maximum load is the basis for calculating the compressive strength of concrete materials according to the formula:(1)$$f_{c}=\frac{F}{A_{c}}[MPa]$$
where:

f_c_—compressive strength [MPa],

A_c_—area of the cross-sectional area of the specimen on which the compressive force acts [mm^2^],

F—maximum load [N].

## 3. Results

As a preliminary test, heating tests were conducted on foamed geopolymer panels (5 cm) glued with high-temperature flexible adhesive to a 4 mm thick steel sheet (Figure 6 and Figure 7). Trial tests were carried out with an oxy-acetylene torch. The density of foamed geopolymer panels was 350 kg/m^3^. The temperature on the outer surface of the steel sheet was measured with a pyrometer. As a result of the test, it was found that when the surface of the foamed geopolymer was heated for 2 h, the outer surface of the steel sheet was heated to a maximum temperature of 360 °C. Figure 8 also shows the structure (SEM) of the foamed geopolymer containing microspheres (this is a zoomed-in image of Figure 6 (right)).

Tests conducted with an acetylene-oxygen torch allowed us to conclude that foamed geopolymers can be used in welding processes where very high temperatures are involved. Such materials can insulate workstations in welding, plasma surfacing, etc. These are materials that can effectively protect steel structures and prevent them from heating to temperatures that could threaten the stability of such a structure. For this type of geopolymer application, it is very important that the chemical composition of the activators be properly selected so that the most relevant molar ratios are appropriate. The dimensional stability and high temperature resistance of a geopolymer depend on the composition of the binder, including the Si/Al ratio, alkali content, and liquid/solid ratio, among others [11]. Alkali content has a dual effect. On the one hand, a high alkali content accelerates the activation reaction and produces a material with high initial strength. On the other hand, alkali content must be limited to ensure the thermal stability of the material. A high alkali content in a composite exposed to high temperatures accelerates the melting of the material. Another important factor is the SiO_2_/Al_2_O_3_ ratio. The most desirable ratio is between 2 and 4—in this range, the initial strength of the material increases and thermo-stable zeolites are formed. Higher ratios hinder the formation of these structures and promote sintering. At lower ratios, excessive zeolite formation can occur at high temperatures, increasing internal stress in the material [36].

Presented below are the results of fire resistance and insulation capability testing of a 160 mm thick foamed geopolymer. The tests were performed in a special chamber. Mini fire-jet tests were conducted, which consisted of subjecting the geopolymer sheet to a stream of fire from a propane torch. A steel plate was attached to the geopolymer plate using a heat-resistant adhesive. Measurements were taken of the temperatures on the side of the flame action and on the outside (the surface of the sheet). The results are presented in the form of a graph in Figure 9.

The tests presented above using foamed geopolymers confirm their very high temperature resistance as well as their resistance to the erosive effects of fire and their ability to provide temperature insulation. The results of testing solid geopolymers for use in welding processes are presented below. The following tables show the results of testing the thermal conductivity coefficient of solid geopolymers (Table 2), as well as the results of testing their compressive strength (Table 3) and flexural strength (Table 4).

Temperature resistance tests were carried out by exposing the geopolymer panels to 800 °C and evaluating whether cracks and other characteristic features appeared on the surfaces. An example comparison of geopolymer panels before and after temperature treatment is shown in Figure 10.

Fire reaction tests were also carried out according to PN-EN ISO 1716:2018-08: Tests of reaction to fire of products—Determination of gross heat of combustion (calorific value) [37] and PN-EN ISO 1182:2020-12: Tests of reaction to fire of construction products—Non-combustibility tests [38]. The results of these tests are shown in the table below (Table 5).

In a further stage of research, to confirm the suitability of geopolymer materials for welding applications, tests were conducted using geopolymer molds for thermite welding of railroad tracks. Molds for thermite welding were made from geopolymers by casting them into a mold specially made for this purpose. The method of obtaining the geopolymer molds was described in Chapter 2 and shown in Figure 3, Figure 4 and Figure 5. The tests were carried out as follows: to begin with, the rails were set up, and the amount of welding clearance, the straightness of the rails in the horizontal plane, and the amount of elevation of the ends of the rails were checked using a ruler and a gap gauge. A stand was installed after the ends of the rails were set up. The stand was used to mount the molds, torch, and crucible. The next step was to check the condition of the molds; then, they were placed in the clamps and mounted to the rack. The gaps between the molds and the rail and the clamps were sealed with sealing sand. Correctly prepared molds are shown in Figure 11.

The next procedure was to prepare a disposable crucible to be placed in a central position, directly on the molds. The burner was placed on a tripod and heating of the ends of the rails to 1000 °C for two minutes was started. The next step was to set the crucible directly on the mold and ignite the thermite portion with a flash igniter. As a result of the reaction, drainage of the liquid melt occurred spontaneously from the crucible into the mold. The melt then began to fill the mold, and the slag was poured into containers that were attached to the mold covers. Selected visualizations from the performed test are shown below in Figure 12.

Figure 13 shows the results from the thermal imaging camera which illustrate the temperatures reached during the thermite welding test. It can be seen that the geopolymer molds heated up to temperatures identical to those of the steel that was welded. The geopolymer cladding was surface-heated to temperatures above 600 °C. It should be assumed that the internal temperature was much higher. The appearance of the samples after the welding process and cooling of the molds, shown in Figure 14, confirms the geopolymer’s very high resistance to high temperatures. No cracking of the molds was observed. The geopolymer molds were stable throughout the test and also after the entire stand cooled down.

## 4. Discussion

There is particular interest in geopolymer materials in the context of their use as insulation materials [39]. Unfortunately, it is often the case that the process of foaming and controlling geopolymer mass with blowing agents is difficult and there are problems with the reproducibility of the results [40]. Problems also occur with various additives that improve thermal insulation properties [41]. There can also be problems with accurately determining the thermal conductivity coefficient and measurements by other methods may often be necessary in addition to the use of a slab apparatus [42,43]. Equally important may be the use of unfoamed geopolymers for technological processes where much greater resistance to high temperatures is required. The presented innovative idea of using fly ash-based geopolymers for the manufacture of molds for termite welding of railroad rails shows that this is possible and can bring many benefits. We are talking, for example, about ecological benefits, since the molds are destroyed after welding and the production of geopolymers from waste means that the natural components used in the production of such ceramic molds are not wasted. Preliminary tests have confirmed the feasibility of using geopolymers to produce molds for thermite welding. The molds passed the entire test successfully and were not damaged.

## 5. Conclusions

Geopolymer materials, due to their attractive properties in terms of temperature resistance, can find a wide range of applications in industry, mainly where materials that can withstand high thermal stresses are needed [44,45]. To date, few researchers have attempted to use geopolymer materials in metal joining technology. This article presents preliminary test results related to the use of geopolymer materials for the thermite bonding of steel railroad rails.

The studies and thermite welding tests carried out using geopolymer molds allowed us to formulate the following conclusions:Geopolymers have a very high thermal resistance and do not degrade at temperatures above 1000 °C;They can be an interesting alternative to the current materials used for thermite welding molds;Several additional studies that can completely confirm the possibility of producing thermite welding molds from geopolymers should be carried out. The problems that were observed when trying to weld rails using geopolymer molds were due to the channels in the molds being too small. The reason for this was that a two-part mold was used to make the geopolymer material and the target geopolymer mold had to be ground and glued. As a result of these procedures, in the final mold produced, the cross-sections of the holes were altered. It is necessary to make a professional casting mold for further research.

This work is currently being continued by the article’s authors and the results will be presented in future articles. Efforts have also been made to use geopolymer materials in other metal joining technologies as well.

The main difference between the traditional approach and the one we propose is the use of geopolymers instead of traditional ceramic forms. Geopolymers can be made from industrial waste, do not have to be fired at high temperatures like ceramic molds, etc. The use of geopolymers for this type of application can bring several benefits, both environmental (energy savings, lower CO_2_ emissions, waste utilization, savings in the consumption of natural raw materials) and technological—geopolymers can be designed and manufactured in a variant resistant to temperatures of several thousand degrees Celsius.

Geopolymer materials are an extremely attractive and perceptive material that may be used in many areas in the near future. The results presented in this work confirm that it is possible to use them in a way that has not been practiced before. Great hopes for geopolymers are seen in their use in additive technologies, such as 3D printing [46], as well as in advanced protective and functional coatings, including as carriers for photocatalytic materials [47]. Regardless of the purpose of geopolymer materials, a very important issue is the appropriate method of their synthesis and hardening [48]. Undoubtedly, this is a technology that requires an appropriate technological regime.

## Figures and Tables

**Figure 1 materials-16-07035-f001:**
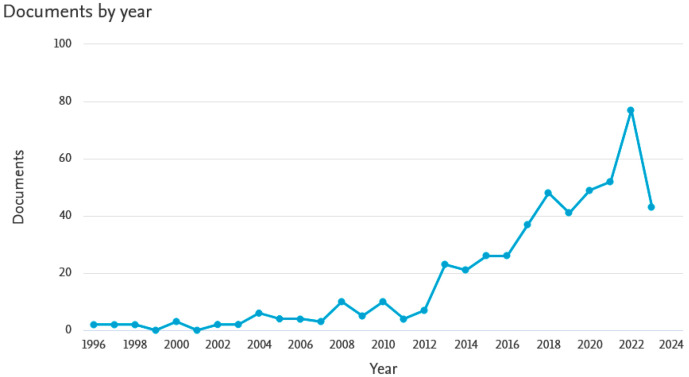
Number of publications on fire resistance of geopolymers according to SCOPUS database.

**Figure 2 materials-16-07035-f002:**
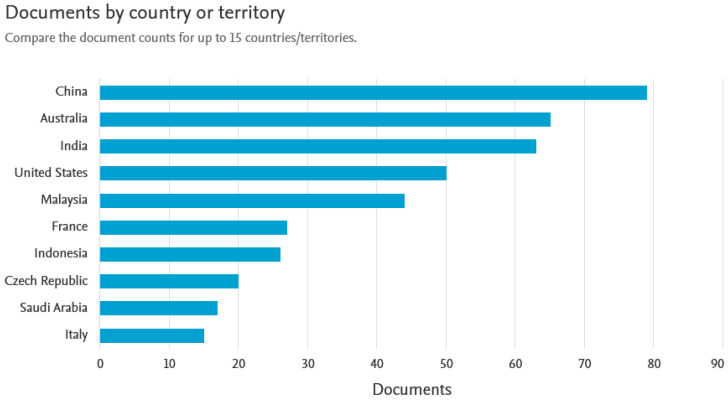
Countries where scientists most often undertake research on geopolymers as fireproofing.

**Figure 3 materials-16-07035-f003:**
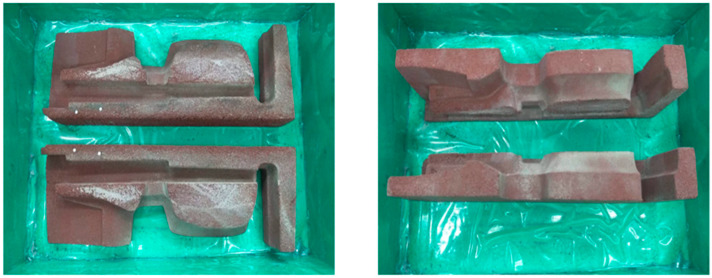
Prepared models for making a thermite welding mold from geopolymers.

**Figure 4 materials-16-07035-f004:**
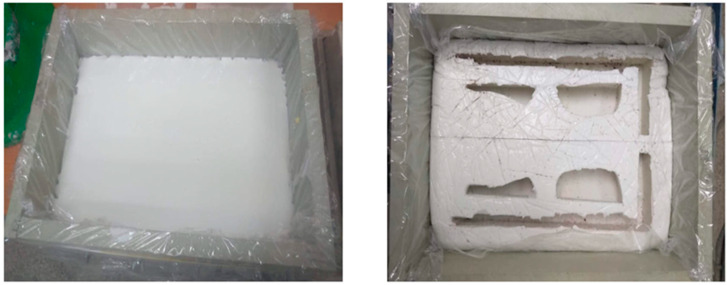
Models (pattern) of the mold flooded with silicone and the finished mold.

**Figure 5 materials-16-07035-f005:**
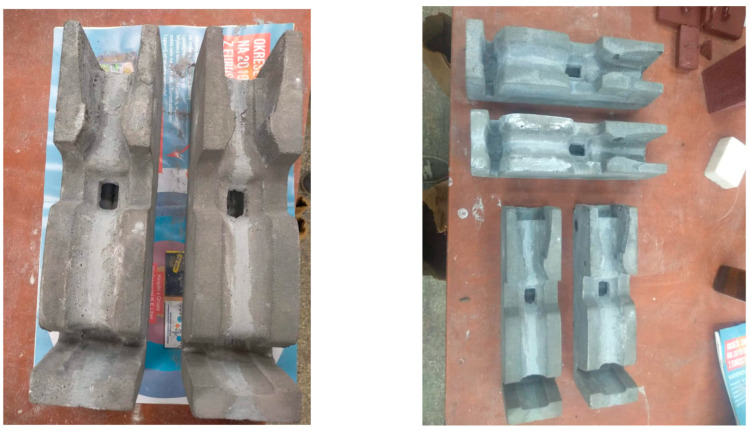
Thermite welding molds made of geopolymers by casting and bonding.

**Figure 6 materials-16-07035-f006:**
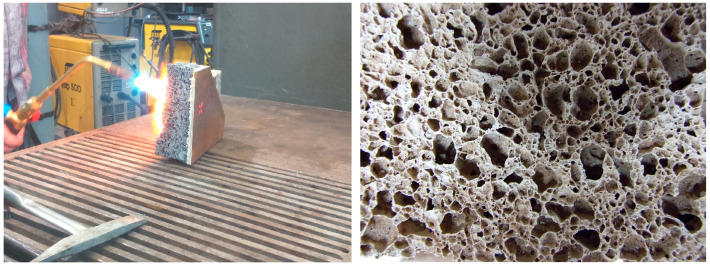
Oxy-acetylene torch testing of a coating (sample fragment—preliminary tests) of foamed geopolymer applied to a steel sheet (**left**) and the microstructure of foamed geopolymer (**right**).

**Figure 7 materials-16-07035-f007:**
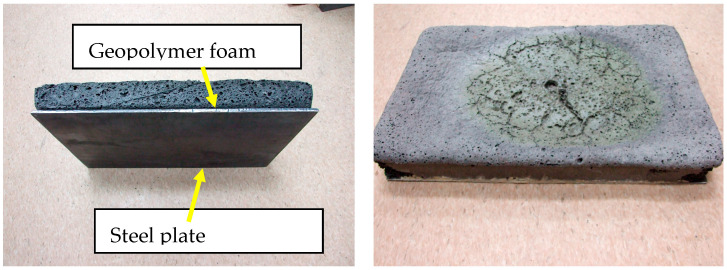
A foamed geopolymer panel attached to a steel sheet (**left**) and the appearance of the surface after an acetylene-oxygen torch test (**right**).

**Figure 8 materials-16-07035-f008:**
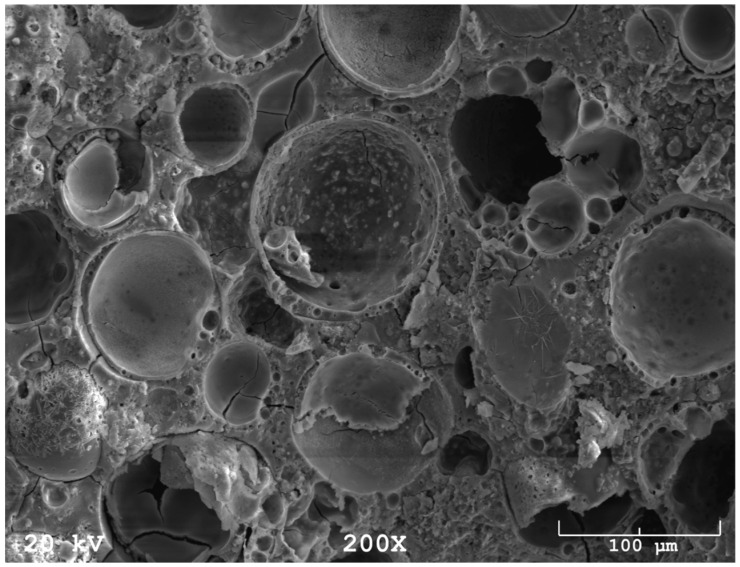
Structure (SEM) of foamed geopolymer containing microspheres.

**Figure 9 materials-16-07035-f009:**
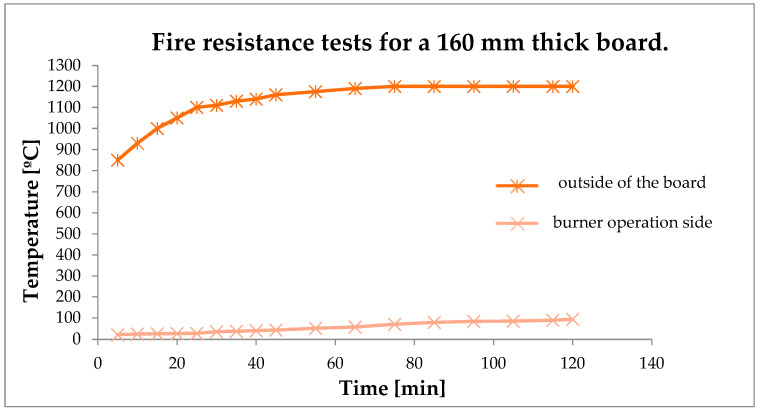
Fire resistance test results for a 160 mm thick board (beige color—temperature measured on the outside of the board; brown color—temperature on the side of the burner operation).

**Figure 10 materials-16-07035-f010:**
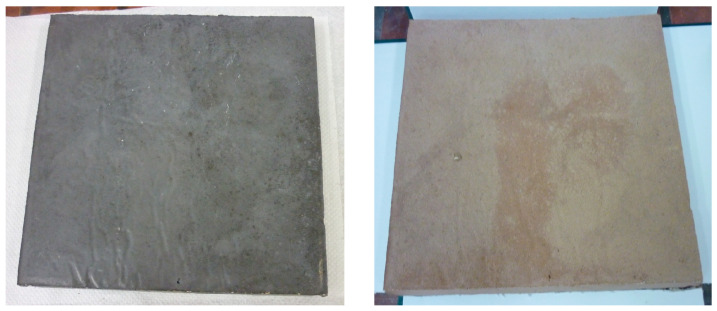
Geopolymer plate before thermal treatment at 800 °C (**left**); geopolymer plate after thermal treatment at 800 °C—surface with no signs of mesh cracks after high-temperature treatment (**right**). (Samples measuring 25 × 25 × 2 cm.)

**Figure 11 materials-16-07035-f011:**
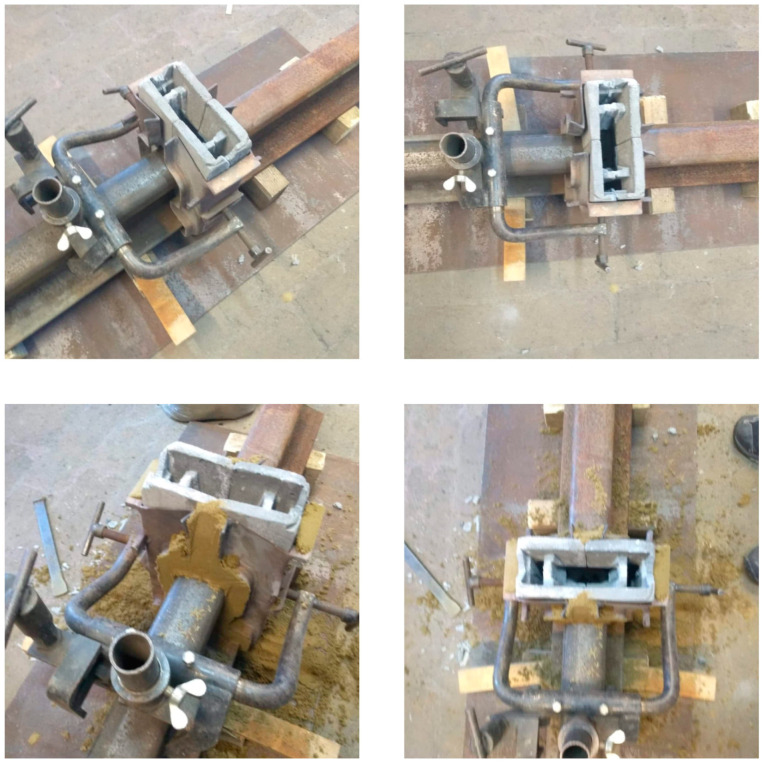
Stand prepared for thermite welding using specially prepared geopolymer molds.

**Figure 12 materials-16-07035-f012:**
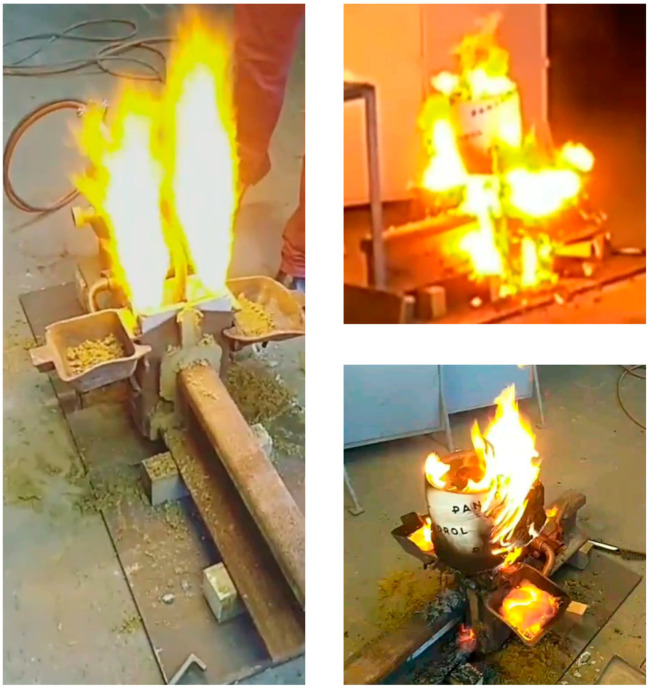
Visualization of the conducted test of thermite welding of railroad rails using geopolymer molds.

**Figure 13 materials-16-07035-f013:**
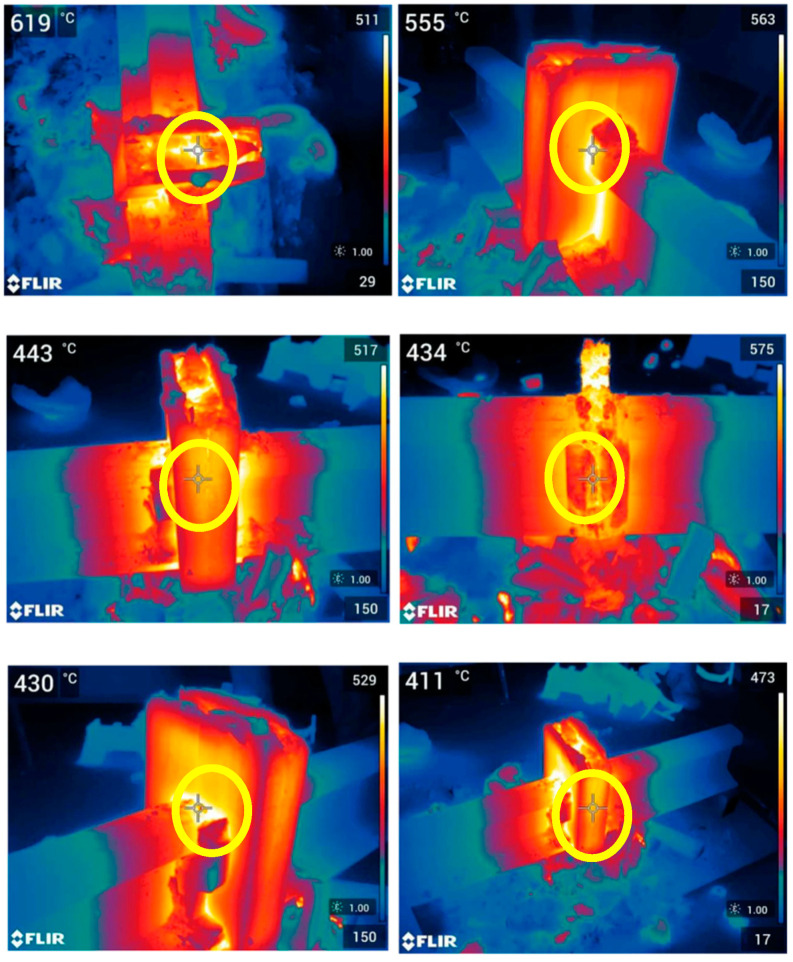
Images from a thermal imaging camera showing the temperature distribution during a rail thermite welding test using geopolymer molds.

**Figure 14 materials-16-07035-f014:**
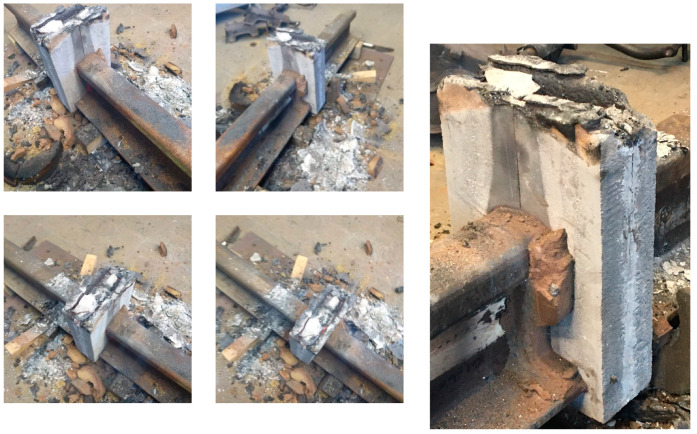
Photos after the test using geopolymer molds.

**Table 1 materials-16-07035-t001:** Oxide analysis of fly ash.

Precursor	Oxide Composition (wt%)							
	SiO_2_	TiO_2_	Fe_2_O_3_	Al_2_O_3_	CaO	MgO	K_2_O	Na_2_O
Fly ash	55.9	1.09	5.92	23.49	2.72	2.61	3.55	0.59

**Table 2 materials-16-07035-t002:** Thermal conductivity of solid geopolymers.

ID	Mean Temp. (°C)	Delta Temp. (K)	Thermal Conductivity (W/m × K)	U_TC (%)	Thermal Resistance (m^2^ × K/W)	U___TR (%)	Temp. Gradient (K/m)	Duration (hh:mm:ss)	Load Pressure (kPa)
1	9.7	4.2	0.88978	1.9	0.0239	1.7	197.61	00:54:40	2.8

**Table 3 materials-16-07035-t003:** Compressive strength of solid geopolymers.

ID	S (mm^2^)	F (kN)	Rm (MPa)
1	2480.59	80.77	32.56
	2582.16	86.11	33.35
	2559.24	84.17	32.89
	2466.80	76.86	31.16

**Table 4 materials-16-07035-t004:** Flexural strength of solid geopolymers.

ID	S (mm^2^)	F (kN)	R (MPa)
1	570.69	3.38	5.91
	563.12	3.55	6.31

**Table 5 materials-16-07035-t005:** Fire reaction test results.

Features Examined	Value
Heat of combustion (mJ/kg)	−0.44
Sample weight before testing (g)	98.34
Sample weight after testing (g)	93.45
Weight loss (%)	4.97
Weight loss (g)	4.89
Duration of the test (s)	1800
Duration of flame combustion (s)	0
Initial furnace temperature (°C)	751.00
Final furnace temperature (°C)	767.66
Final surface temperature of the sample (°C)	777.97
Maximum furnace temperature (°C)	768.31
Maximum surface temperature of the sample (°C)	778.65
Furnace temperature increment ΔT (°C)	0.66
Increase in sample surface temperature ΔTs (°C)	0.68

## Data Availability

Not applicable.

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
