# Peer review of "Possibilities of Using Geopolymers in Welding Processes and Protection against High Temperatures"

_materials, 2023, doi:10.3390/ma16217035_

Round 1
Reviewer 1 Report (New Reviewer)
Comments and Suggestions for Authors
materials-2680317
Possibilities of using geopolymers in welding processes and protection against high temperatures
After careful evaluation, I have concluded that the above-mentioned manuscript requires major revisions
Comments
1. In lines 88 and 89, the authors' use of the word "thanks" seems unprofessional. This platform is not meant for expressing gratitude; providing a reference to support their statement would be more appropriate.
2. What was the specific composition of the aqueous solution used in the preparation of the geopolymers for the rail thermite welding tests?
3. What were the specific properties of the technical sodium hydroxide flakes and the aqueous solution of sodium silicate R-145 used in the geopolymer masses?
4. Why was distilled water not utilized in the process, and what type of water was used instead?
5. How was the alkaline solution prepared, and what was the specific method used for achieving equilibrium in concentrations and temperature?
6. What specific process was followed for mixing the solid ingredients (fly ash and sand) with the alkaline solution, and what equipment was used for this purpose?
7. Certainly, here are five questions based on the provided text regarding the use of geopolymer materials for welding applications:
8. How were the molds for thermite welding created using geopolymers, and what was the specific process involved in casting the geopolymers into the mold?
9. Where can the detailed description of the method for creating geopolymers molds be found, and what specific figures illustrate the process?
10. What were the key parameters that were examined during the conducted tests for the thermite welding of railroad tracks using the geopolymer molds?
11. What was the purpose of installing a stand after setting up the ends of the rails, and how was this stand utilized in the welding process?
12. What was the procedure for checking the condition of the molds before they were mounted in the clamps and on the rack for the welding process?
13. Certainly, here are five questions based on the provided text regarding the use of geopolymer materials as insulation and in the manufacturing of molds for thermite welding:
14. What are some common difficulties encountered in the process of foaming and controlling geopolymer mass with blowing agents, and what issues arise with the reproducibility of results?
15. What are the challenges associated with the use of various additives to enhance thermal insulation properties in geopolymer materials?
16. How is the determination of the thermal conductivity coefficient problematic, and why might alternative measurement methods be necessary alongside the use of a slab apparatus?
17. Aside from their application as insulation materials, what other potential uses are suggested for geopolymers, particularly in processes requiring high temperature resistance?
18. What ecological benefits are associated with the use of fly ash-based geopolymers in the production of molds for thermite welding, and how do these benefits relate to the efficient utilization of natural components and waste materials?
19. After thoroughly reviewing the entire manuscript, it has come to my attention that the concept of geopolymerization mechanism is missing. To enhance the comprehensibility regarding the formation of the tetrahedral structure and the mechanism behind strength development, I would recommend the authors consider incorporating a dedicated section on this topic. This addition would significantly contribute to a better understanding of the overall manuscript. Below are few articles that can provide insight into this concept.
doi.org/10.1016/j.conbuildmat.2023.132869 ; doi.org/10.3390/coatings12091348; doi.org/10.1016/j.jmrt.2023.02.088
Comments on the Quality of English LanguageMinor editing of English language required.
Author Response
Dear Reviewer,
Thank you very much for all your comments. We tried to take into account all suggestions. Please read the revised version of the manuscript.
We ask for your understanding. Our goal was mainly to present selected properties of the geopolymer in the context of its fire (temperature) resistance and to present only the concept and preliminary test results of the use of geopolymers in thermite welding processes of railway rails. We are aware that the results presented are not at a high level, but they are only preliminary test results. We wanted the scientific community to become familiar with the general concept of the possibility of using geopolymers for this purpose. We are currently working on expanding the scope of the research and would like to describe specific results in a future article. Our superiors agree to continue the work but believe that we should present the current results in a scientific journal. We kindly ask you to understand our situation and accept our manuscript.

Reviewer 2 Report (Previous Reviewer 2)
Comments and Suggestions for Authors
The authors answered all reviewers' comments and the quality of the paper increased.
Author Response
Dear Reviewer,
Thank you very much for accepting our manuscript. We made an effort to improve a few more things and now it looks better than before.
Reviewer 3 Report (Previous Reviewer 3)
Comments and Suggestions for Authors
Accept
Author Response
Dear Reviewer,
Thank you very much for accepting our manuscript. We made an effort to improve a few more things and now it looks better than before.
Reviewer 4 Report (Previous Reviewer 4)
Comments and Suggestions for Authors
Dear Authors
The work looks poor and not properly presented such as although author claims burning geopolymer and its correlation with chemical composition such Si/Al ratio. How do the author prove that? Is there any EDX analysis ? We couldnot predict the data as hypothesis. Fig. 11, 12 and 14 are reductant, what could we able to interpret from these images ? What Fig. 15 says?
Fig. 13 shows temperature distribution in geopolymer burning stage at different location. However its poorly presented, it should improve. There is cross which mark the place where temperature is located ?
Almost all the picture is without scale. Such as Fig. 10 there is no scale. how authors determine thermal conductivity, compressive strength and flexural modulus of geopolymers ?
There is lacking of concept, presentation. In the present stage the article is not ready for publication.
Author Response
Dear reviewer,
Thank you very much for all your comments. We have improved our article significantly. Please read the revised version of the manuscript. We ask for your understanding. Our goal was mainly to present selected properties of the geopolymer in the context of its fire (temperature) resistance and to present only the concept and preliminary test results of the use of geopolymers in thermite welding processes of railway rails. We are aware that the results presented are not at a high level, but they are only preliminary test results. We wanted the scientific community to become familiar with the general concept of the possibility of using geopolymers for this purpose. We are currently working on expanding the scope of the research and would like to describe specific results in a future article. Our superiors agree to continue the work but believe that we should present the current results in a scientific journal. We kindly ask you to understand our situation and accept our manuscript.

Round 2
Reviewer 1 Report (New Reviewer)
Comments and Suggestions for Authors
ID: materials-2680317R1
Title: Possibilities of using geopolymers in welding processes and protection against high temperatures
Comments: The authors have taken the time to carefully address all of the comments and suggestions provided by me previously. Their efforts in revising the manuscript have been thorough and have resulted in significant improvements. As a result, I am pleased to announce that the revised version of the manuscript meets the required standards for acceptance. I appreciate the authors' commitment to enhancing the quality of their work and their willingness to implement the required changes. Therefore, it is my pleasure to inform the authors that their revised manuscript has been deemed suitable for acceptance.
Reviewer 4 Report (Previous Reviewer 4)
Comments and Suggestions for Authors
Author has done necessary modification. Modified meets the criteria.
Accept
This manuscript is a resubmission of an earlier submission. The following is a list of the peer review reports and author responses from that submission.
Round 1
Reviewer 1 Report
Comments and Suggestions for Authors
16/August/2023
Review of the manuscript 2570232
The title is “Possibilities of using geopolymers in welding processes and protection against high temperatures”
Authors introduce new application of geopolymers in mold of thermite welding. I think this paper is meaningful from the viewpoint of SDG’s and safe railroad operation. I have some comments as follows.
1. Abstract, line 21: Document of “welding of rail rails” should be “welding of railroad rails”.
2. 3. Results: Authors have to explain about Fig.6 and Fig.7 in the text. These figures are not cited in the text.
3. Page 4, line138, Table 1.: Authors mentioned main oxide compositions in the fly ash. I am concerned about the elements of phosphorus(P), sulfur(S) and carbon(C) that promote cracking of weld metal. Are these elements enough low?
4. Fig. 6, Fig. 7: Insufficient description of figures. Please mark arrows and explain where the torch, the geopolymer and the steel are.
5. Page 9, line 298-300: Authors mainly focused to molds temperature. However, cooling rate of steel rails that temperature-time diagram is more important from the weld strength and defect points of view. Authors have enough data, like Fig.10, to draw the diagram.
6. Page 9, line 302: Authors say “No cracking of the molds “. I think more important things is “No cracking of the rail welded”. Please prove with macro photograph that are no cracks in the weld. This is an essential observation that I always do with weld joint.
These are my comments.

Reviewer 2 Report
Comments and Suggestions for Authors
REVIEWER SUGGESTED GENERAL COMMENTS:
The reviewer congratulates the authors for their ability to present am exploratory investigation, but regrets to inform that this publication is a preliminary investigation and additional studies should be carried out. According to the authors, this investigation is going on and they should do an effort to include deterministic results.
The proposed manuscript CANNOT BE ACCEPTED, as it is.
THE REVIEWER SUGGESTS SOME GENERAL COMMENTS TO THE DOCUMENT, IN PARTICULAR:
Authors should also address any concerns that potential readers may have related with this manuscript:
1- PAGE 7: LINE 224: Tests conducted with an acetylene-oxygen torch allow the authors to conclude that foamed geopolymers can be used in welding processes. How? Did authors measure the exposed temperature when exposed to the torch? Please give evidences of this capacity and explain the criterion used.
2- PAGE 7: LINE 241: Figure 6 and 7 are not cited before Figure 8. Please check it.
3- PAGE 11: LINE 359: Authors made Preliminary tests without analysis (only empirical methods are presented). No data is presented. Lack of scientific background.
4- PAGE 11: LINE 375: Conclusion 1 is not supported by any measurement.
Comments on the Quality of English LanguageNo comment.
Reviewer 3 Report
Comments and Suggestions for Authors
An interesting research topic, this manuscript describes using geopolymers as welding molds. Research brings new insights into making insulating materials and welding methods. Experimental results show the effectiveness of the method, but lack of systematic microscopic analysis. The correlation between the experiment and the analysis of the article is weak. I suggest making major revisions before considering whether to accept it.
My detailed review follows:
Line 139: Please confirm whether this is XRD or XRF.
Line 144-145. Why is the stirring time chosen to be 10 minutes? Is it done according to any standard?
What is the purpose of adding 3% hydrogen peroxide H2O2 to Line 145? The author needs to explain.
The bonding process of the two-part mold of Line 169 needs to be introduced in detail, which is very important for the workability of the mold.
The Line 209 authors used foamed geopolymers bonded to steel plates. What is the thickness of the bonded geopolymer? Does the thickness affect the results of the cast?
Line 224-228: The author said, "foamed geopolymers can be used in welding processes where very high temperatures above 3000 °C are used." How does the author get this conclusion from the experimental results? The correlation between the author's experimental results and conclusions is too poor. I suggest the author reorganize the language in this section.
In addition, the operation of the experiment should be organized in part of Materials and Methods.
Fig. 11: Although the author shows mold photos from different angles, these photos are not enough for reference. Please provide high-quality pictures or close-up shots of the mold's condition.
Fig. 12: The content of the picture is difficult for readers to understand. The picture needs to be annotated.
In this manuscript, the authors used geopolymers but did not detail the benefits of this practice. The difference between this approach and the traditional approach needs to be mentioned.
Comments on the Quality of English LanguageMinor editing of English language required
Reviewer 4 Report
Comments and Suggestions for Authors
Dear Authors
Article begins with good aim , however the content is not significant according to scientific soundness. Table 1 and fig. 10 sounds scientific however other figures are visualization, without any data, or molds image without any scale.
Although the article starts with good aim, the final achievement was lower.
Authors could explain mold design in scheme with proper dimension and scheme. Material lost or high temperature resistance should plotted with data.
In this stage, the article is not suitable for publication.
Sorry , authors could follow the articles for future
Correlation of microstructure and mechanical properties of various fabric reinforced geo-polymer composites after exposure to elevated temperature
Ceramics International 41 (9), 12115-12129
Improved mechanical properties of various fabric-reinforced geocomposite at elevated temperature
Jom 67, 1478-1485
Thermal Characterization of Metakaolin-Based Geopolymer
JOM, 1-5